# Pancreatic Adenocarcinoma Up-Regulated Factor (PAUF) Transforms Human Monocytes into Alternative M2 Macrophages with Immunosuppressive Action

**DOI:** 10.3390/ijms252111545

**Published:** 2024-10-27

**Authors:** Yeon Jeong Kim, Sitansu Sekhar Nanda, Fen Jiang, Seung Yeon Pyo, Jin-Yeong Han, Sang Seok Koh, Tae Heung Kang

**Affiliations:** 1Prestige Biopharma IDC, Busan 46726, Republic of Korea; yeonjeong@prestigebio.com (Y.J.K.); sitansusekhar.nanda@prestigebio.com (S.S.N.); fen.jiang@prestigebio.com (F.J.); seungyeon.pyo@prestigebio.com (S.Y.P.); sangseok.koh@prestigebio.com (S.S.K.); 2Department of Biomedical Sciences, Dong-A University, Busan 49315, Republic of Korea; 3Department of Laboratory Medicine, College of Medicine, Dong-A University, Busan 49201, Republic of Korea; jyhan@dau.ac.kr

**Keywords:** TME, PAUF, TAM, TLR, M2 macrophages

## Abstract

Tumor-associated macrophages (TAMs) in the tumor microenvironment (TME) promote immune evasion, cancer cell proliferation, and metastasis. Ongoing research is focused on finding ways to prevent tumor growth by inhibiting TAM polarization, which has shown a correlation with unfavorable prognosis in clinical studies. Pancreatic adenocarcinoma up-regulated factor (PAUF) is a protein secreted from pancreatic cancer (PC) and acts as a TME modulator that affects the TME by acting on not only cancer cells but also stromal cells and immune cells. Tumor cells can evade the immune system by PAUF binding to Toll-like receptor (TLR) in monocytes, as this research shows. In this study, the examination centered around the recruitment of human monocytes by PAUF and the subsequent differentiation into macrophages. In an in vitro chemotaxis assay, PAUF induced chemotactic migration of TLR2-mediated monocytes. In addition, PAUF induced differentiation of monocytes into M2 macrophages, which was verified based on expressing surface markers and cytokines and morphological analysis. The inhibition of T cell proliferation and function was observed in differentiated M2 macrophages. To conclude, these findings indicate that PAUF functions as a promoter of cancer progression by regulating the recruitment and differentiation of macrophages within TMEs, ultimately causing immunosuppression.

## 1. Introduction

Pancreatic ductal adenocarcinoma (PDAC) cells show that significant expression of the novel secreted protein, PAUF, contributes to the progression of PC by modulating the TME through paracrine signaling [1]. For instance, it amplifies the immune-suppressive capabilities of immune cells via TLR-mediated pathways (TLR2 and TLR4) [1]. Cancer progression and metastasis are positively associated with the ongoing interaction between cancer cells and stromal cells in the TME [2]. Loss of epithelial cell function and abnormal regulation of the functions of stromal cells surrounding cancer cells are important processes for tumorigenesis [3,4,5]. The epithelial cells induce fibrosis in the extracellular matrix and recruitment of stromal cells [6,7,8]. Therefore, it is necessary to understand the molecular pathology underlying the relationship between stromal cell recruitment into the TME and tumor progression. More specifically, the TME in PC is made up of diverse stromal cells like TAMs, myeloid derived suppressor cells (MDSCs), cancer associated fibroblasts (CAFs) and Tie2 expressing monocytes (TEMs), which contribute to the aggressive nature of the TME [9].

Macrophages, derived from the bone marrow, are crucial innate immune cells [9]. Upon encountering molecules from infected cells or foreign substances, immature monocytes leave the bone marrow through the bloodstream and undergo activation, ultimately developing into mature macrophages [10]. Two types of mature macrophages exist depending on the stimulatory signal: M1 macrophages, or classically activated macrophages, and M2 macrophages, or alternatively activated macrophages [11,12]. TAMs are a specific type of macrophages that are found in the TME, express cytokines and chemokines like M2 macrophages, and promote cancer growth and survival [13]. TAMs promote the proliferation of T-helper 2 (Th2) and not Th1 cells by producing pro-inflammatory cytokines and activate regulatory T cells to induce immune tolerance [14,15]. TAMs promote angiogenesis by producing anti-inflammatory cytokines and support the invasive and metastatic abilities of tumor cells [13]. TAMs promote tumor growth by inhibiting apoptosis induced by anti-cancer drugs [14]. The presence of TAMs in tumors leads to tumor progression via the secretion of growth factors, cytokines, and other signaling molecules. They stimulate the growth of Th2 cells, trigger Tregs to promote immune tolerance, and hinder Th1-mediated immune responses. The production of anti-inflammatory cytokines by TAMs promotes angiogenesis and enhances the invasive and metastatic capabilities of tumor cells. TAMs have been studied as possible targets for therapy to inhibit tumor growth and spread due to their significant role in the tumor microenvironment.

These results suggest that the complex interaction between immune and tumor cells in the TME intricately regulates tumor progression. Recent clinical studies have reported that TAMs in the TME are associated with poor prognosis in several cancers, such as PC [16], bladder cancer [17], gastric cancer [18], and breast cancer [19]. Inhibiting monocyte differentiation into M2 macrophages is an active area of research for developing therapies that can effectively impede tumor growth [20].

PAUF is involved in the functional regulation of immune and cancer cells in the TME. PAUF induces monocyte activation through TLRs to promote tumor growth and escape from immune surveillance [21]. Koh et al. [22] found that PAUF enhances the immunosuppressive ability of MDSCs, leading to increased production of arginase, nitric oxide (NO), and reactive oxygen species (ROS) via the TLR4-mediated the mitogen-activated protein kinase/extracellular signal-regulated kinase (MAPK/ERK) pathway. A PAUF-neutralizing antibody was used to further confirm these findings in a mouse model of PC and MDSCs from patients with PC. To explore their clinical relevance, monocytes derived from human peripheral blood mononuclear cells (PBMCs) were utilized. This study focused on examining how PAUF influences monocyte chemotaxis and their subsequent differentiation into macrophages. This was evaluated by analyzing surface markers and cytokine expressions. Additionally, it examined how PAUF interacts with immune cells, as it is believed to function as a modulator in the TME.

## 2. Results and Discussion

### 2.1. PAUF Increases TLR-Mediated Chemotactic Migration of Monocytes

Primary human monocytes were used to investigate the relationship between PAUF and TAMs. Magnetic beads were used to isolate cluster of differentiation 14 (CD14)-positive monocytes from PBMCs obtained from healthy donor blood. The purity of the isolated monocytes was assessed through flow cytometric analysis (Figure 1A). Microscopic examination (Figure 1B) of CD14, CD11c and CD3 pan-surface markers revealed expression of monocyte, dendritic cell, and lymphocyte. CD14-positive monocytes had a purity of more than 95%.

In order to study the recruitment of monocytes, the chemotactic effect of PAUF was assessed through a transwell assay (Figure 2A). The isolated monocytes were placed in the upper chamber and media with different PAUF concentrations were added to the lower chamber to assess monocyte migration. In comparison to the control group, monocyte migration was enhanced in a concentration-dependent manner by PAUF (Figure 2B).

According to a previous study, PAUF acts as a paracrine factor and binds to TLR2 in monocytes, which might inhibit nuclear factor kappa B (NF-κB) signaling due to TLR-mediated immune cell activation, facilitating escape of tumor cells from immune surveillance [23]. Blocking TLR2 inhibitors halted the ability of PAUF to attract monocytes and induce their chemotactic migration (Figure 2C). Similarly, treatment with Pam3Cys (TLR2 ligand) also inhibited the chemotactic effect of PAUF, showing that PAUF regulates monocyte activation and recruitment by mediating TLR2 (Figure 2C). The results of the research demonstrated that elevated PAUF levels in the TME facilitate the attraction of monocytes and stimulate a tumor-promoting immune response.

### 2.2. PAUF Induces Monocyte Differentiation into M2 Macrophages

This investigation focused on the transformation of monocytes into M2 macrophages when exposed to PAUF, aiming to comprehend the immunosuppressive mechanism of PAUF in the TME (Figure 3). It consistently transformed monocytes into monocyte-derived macrophages (MDMs) in vitro by employing granulocyte-macrophage colony-stimulating factor (GM-CSF) for a duration of 7 days, resulting in GM-MDMs, or macrophage colony-stimulating factor (M-CSF), resulting in M-MDMs [24]. GM-MDMs, M-MDMs, and PBS-MDMs were used as M1 macrophages, M2 macrophages, and negative controls, respectively (Figure 3A). As shown in Figure 3B, PBS-MDMs (M0; day 7) showed a small and roundish morphology, similar to that of monocytes, whereas GM-MDMs (M1; day 7) showed expanded cytoplasm with roundish cell bodies and many microvilli on the cellular surface, and M-MDMs (M2; day 7) were spindle-shaped, with an elongated cell body and cytoplasmic extensions. This investigation confirmed that macrophages treated with PAUF (PAUF-MDMs) have morphologies similar to that of M2 macrophages (Figure 3A). However, PAUF was not as potent as M-CSF in stimulating monocyte differentiation into M2 macrophages.

M2 macrophages play a role in suppressing parasites, remodeling tissues, and promoting tumor progression. Their characteristics include phagocytic activity, high expression of scavenger and mannose receptors, and arginase activity [25,26,27]. As a consequence, the expression of CD86 and CD163 markers to determine the polarization of differentiated macrophages into M1 and M2 phenotypes was analyzed and a higher expression of CD86 in GM-MDMs was observed (Figure 3B), while M-MDMs and PAUF-MDMs showed a higher expression of CD163 (Figure 3C). The expression of CD206 analyzed a mannose receptor, which is a marker for M2 macrophages, using immunofluorescence analysis. CD206 expression was similar to that observed for CD163 (Figure 3D). Next, the phagocytic activity of M2 macrophages was observed, and PAUF-MDMs using a fluorescent protein–carbohydrate complex were used to verify that M-MDMs and PAUF-MDMs exhibit similar phagocytic activity (Figure 4). These data suggest that PAUF and M-CSF share functional similarity, leading to monocyte differentiation into M2 macrophages. Macrophage polarization analysis was carried out by flow cytometry following treatment of monocytes with control IgG or anti-PAUF antibody. CD163 and CD206 expressions were considerably lower in anti-PAUF antibody-treated macrophages compared to that in PAUF- and control IgG-treated macrophages, suggesting that PAUF directly regulates monocyte differentiation into M2 macrophages (Figure 4). Taken together, these findings suggest that PAUF recruits monocytes into the TME and induces their differentiation into tumor-promoting M2 macrophages, which can be reversed by treatment with an anti-PAUF antibody. Therefore, a PAUF-target antibody represents a novel strategy for regulating immune cells in the TME.

### 2.3. PAUF-MDMs Express Immune-Suppressive Cytokines

The immune response is stimulated by M1 macrophages that possess anti-cancer properties, which secrete inflammatory cytokines including Interleukin (IL)-I1, IL-6, IL-12, and TNF-α [28,29]. In comparison, M2 macrophages produce cytokines that have anti-inflammatory effects, including IL-10 and TGF-β, which have been shown to promote tumor development. Hence, the levels of TNF-α, IL-10 and arginase are examined in cell culture supernatants through ELISA. TNF-α expression was high in GM-MDMs (Figure 4B), while IL-10 expression was high in M-MDMs. PAUF-MDMs exhibited higher IL-10 expression than M-MDMs (Figure 4B). Arginase expression in M2 macrophages was verified through an Arginase Activity Assay Kit, while iNOS and ROS expression in M1 macrophages was confirmed through flow cytometry. ROS and iNOS expressions were high in GM-MDMs, whereas arginase expression was high in M-MDMs and PAUF-MDMs (Figure 4C,C`). Collectively, these data suggest that the function of PAUF-MDMs is like that of M2 macrophages.

### 2.4. PAUF Acts as an Immunosuppressive Factor by Inhibiting T Cell Proliferation

M2 macrophages promote immune evasion of tumor cells through crosstalk with other immune cells. Arginase, which is mainly expressed in M2 macrophages, inhibits the cytotoxic function of T cells [30]. The secretion of IL-10 by M2 macrophages leads to an increase in regulatory T cell recruitment to the TME, subsequently inhibiting T cell activity [31]. In order to observe the impact of PAUF-MDMs on immune cell activity, macrophages with CD4^+^ or CD8^+^ T cells were cultivated. To analyze the proliferative ability of T cells, CFSE-labeled T cells and macrophages were co-cultured for four days and analyzed by flow cytometry. Figure 5 shows that T cells co-cultured with M-MDMs or PAUF-MDMs exhibited significantly inhibited proliferation ability compared to T cells co-cultured with PBS-MDMs or GM-MDMs. The results imply that PAUF-MDMs serve as M2 macrophages, restricting T cell proliferation and potentially facilitating tumor advancement through the inhibition of T cell mediated anti-cancer immune response. Collectively, PAUF recruits monocytes from the blood into the TME, induces their differentiation into TAMs, and inhibits T cell proliferation by secreting specific cytokines, facilitating tumor progression. It indicates that PAUF serves as a catalyst for cancer advancement by controlling macrophage activity and promoting immunosuppression within the TME. Tumor heterogeneity increases the complexity and difficulty of cancer treatment, as a single drug cannot target all cancer cells [32]. Cancer treatments often fail because researchers focus solely on studying the biological function of target molecules within cancer cells. Hence, scientists need to examine the intricate and dynamic interactions between cancer and stromal cells in the TME [33]. The infiltration of pro-cancer stromal cells into the TME positively correlates with cancer progression and metastasis. Different types of cells, including CAFs, TAMs, TEMs, and MDSCs, are part of TME-specific stromal cells. TAMs promote immune evasion in the TME, resulting in cancer cell growth and spread. The poor prognosis of various cancers is linked to the presence of TAMs in the TME. Hence, the development of a treatment strategy that inhibits tumor growth by blocking the polarization of TAMs is being actively investigated [34]. Unlike M1 macrophages, researchers classify different TAMs based on their gene or cytokine expression profiles. IL-4 and IL-13 stimulate differentiation of M2a macrophages, while immune complexes and LPS stimulate M2b macrophages, and glucocorticoids and TGF-β stimulate M2c macrophages [35]. However, there is a difference in factors expressed by each class of macrophages depending on the concentration of the stimulating factor or the method of treatment, and it is difficult to distinguish between the subtypes clearly. Therefore, the precise mechanisms underlying the polarization of TAMs and markers that can differentiate the various subtypes need to be explained [36].

PAUF, a factor that regulates the metastasis of PC, affects both cancer cells and different stromal cells. A previous study revealed the mechanism underlying immunosuppression mediated by PAUF-induced activation of THP-1 and MDSCs [22]. In addition, the group confirmed PAUF induces recruitment of TAMs to the tumor site. Using monocytes isolated from human peripheral blood, the interaction between PAUF and TAMs was analyzed in the present study. These results revealed PAUF recruited monocytes and induced their differentiation into TAMs. PAUF showed functional similarity to M-MDMs and PAUF-MDMs secreted cancer-promoting cytokines. Stimulation by these cytokines leads to the secretion of pro-tumoral cytokines by other macrophages, endothelial cells, and fibroblasts, forming a vicious cycle that facilitates the formation of an aggressive TME. Furthermore, PAUF hinders T cell proliferation in the TME, enabling immune evasion of cancer cells. It indicates that PAUF plays a role in promoting cancer progression by controlling macrophage influx, inducing angiogenesis, metastasis, immunosuppression, and reducing response to anti-cancer drugs. Research on PAUF and TAMs offers valuable information for developing anti-cancer drugs that control macrophage polarization. The study suggests that PAUF-induced M2-like macrophages can maintain antitumor properties, challenging the traditional polarization concept in tumor microenvironments. The impact of PAUF on TAMs was shown in their ability to suppress T cell functions, including CD4+ T cell proliferation and CD8+ T cell activity. The results indicate that PAUF indirectly influences T cells by modulating their functions through TAM-mediated mechanisms in the TME.

## 3. Materials and Methods

### 3.1. Human Blood Samples

Blood was obtained from excess specimens remaining after the examination, which was completed at the department of Laboratory Medicine of Dong-A University Hospital; and the Dong-A Institutional Review Board (IRB: BR-003-02) approved the study. We utilized around 20 extra samples for each experiment. Each sample provided us with 2–3 mL of blood, and we utilized around 50 mL for each experiment. We usually obtained around 1 × 10^8^ PBMCs from the 50 mL blood, which were then used for the experiments.

### 3.2. Cell Culture

The human blood-derived cells, including monocyte and T cell, were cultured in RPMI-1640 media with 10% FBS and 1% penicillin/streptomycin. All cells were incubated at 37 °C in a humidified atmosphere of 5% carbon dioxide (CO_2_).

### 3.3. Isolation of PBMCs

PBMCs were isolated from human blood within 24 h after obtaining the blood specimens. The human blood was diluted 1:1 in PBS -with 2% FBS, and the PBMCs were isolated using density-gradient centrifugation on a Ficoll Paque Plus (1.077g/mL; GE Healthcare, Chicago, IL, USA) and SepMate-50 tubes (Stemcell Technologies, Vancouver, Canada). Centrifugation was performed at room temperature, 1200× *g*, for 10 min. Following isolation, cells underwent three washes in PBS containing 2% FBS. Cells were counted and then used for analysis.

### 3.4. Isolation of Monocytes

Monocytes were isolated by cluster of differentiation 14 (CD14) negative selection with the human Classical Monocyte Isolation Kit, (#130-117-337, Miltenyi Biotec, Bergisch Gladbach, Germany) using the manual method or the automated machine, an auto magnetic cell separations (autoMACS) Pro instrument (Miltenyi Biotec, Bergisch Gladbach, Germany) as described below. For purity analysis, monocytes were stained with CD3-APC (#17-0038-42, Invitrogen, Waltham, MA, USA), CD11c-PE (#555392, BD Biosciences, Franklin Lakes, NJ, USA) and CD14-FITC (#555397, BD Biosciences) and analyzed using a flow cytometer. The purity was >95%.

### 3.5. Using the MACS Separator

After PBMC isolation, cells were re-suspended in PBS buffer, pH 7.2, containing 0.5% bovine serum albumin (BSA), and 2 mM ethylenediamine tetraacetic acid (EDTA); and incubated with FcR Blocking Reagent (10 μL per 10^7^ cells) and Classical Monocyte Biotin-Antibody Cocktail (10 μL per 10^7^ cells) during 5 min at room temperature. Then the additional incubated with Anti-Biotin MicroBeads (20 μL per 10^7^ cells) for 5 min at room temperature. The cell suspension was loaded onto an LS magnetic column (Miltenyi Biotec) placed in the magnetic field of a MACS Separator (MIDIMACS; Miltenyi Biotec, South Korea) and rinsed three times with buffer. At this point, the unlabeled cells, representing the enriched CD14^+^ monocytes were eluted.

### 3.6. Using the AutoMACS Pro Separator

CD14^+^ monocyte isolation was performed by autoMACS Pro Separator. The isolation process followed the instructions provided by the manufacturer. Briefly, the PBMCs and collection tubes were placed into the Chill Rack and the Deplete2 program was selected to start the separation. After separation, the negative fraction (CD14^+^ monocytes) was collected from row B of the tube rack.

### 3.7. Isolation of T Cells

T cells were isolated from PBMCs by negative selection using a human CD4^+^ T Cell Isolation Kit (#130-091-155, Miltenyi Biotec) and a human CD8^+^ T Cell Isolation Kit (#130-096-495, Miltenyi Biotec). Isolation of T cells was performed by autoMACS Pro Separator. Both cells were used by collecting the negative fraction using the Deplete program.

### 3.8. Differentiation of Monocytes into Macrophages

For macrophage differentiation, freshly isolated monocytes were seeded in a 60 × 15 mm cell culture dish (SPL Life Sciences, Gyeonggi-do, Korea) at an appropriate concentration, such as 4.375 × 10^5^ monocytes/well, in 3 mL RPMI-1640 supplemented with 10% FBS and 1% penicillin/streptomycin. Human CD14^+^ monocytes were cultured with recombinant human GM-CSF (granulocyte-macrophage colony-stimulating factor) (GM-CSF; 50 ng/mL, #300-03, PeproTech, Cranbury, NJ, USA) or recombinant human M-CSF (macrophage colony-stimulating factor) (50 ng/mL, #300-23, PeproTech, Cranbury, NJ, USA) for M1 and M2 macrophage polarization, respectively. To observe the differentiation of macrophages by PAUF, PAUF (1.0 μg/mL) was treated, and PBS was treated as a negative control. The cells were cultured in a 5% CO_2_ incubator at 37 °C for 7 days.

### 3.9. Chemotactic Migration Assay

Chemotactic migration was analyzed in transwell (Corning Costar, Cambridge, MA, USA) membrane. The upper chambers were filled with cell suspensions (3 × 10^5^ cells/well) in serum-free RPMI-1640 medium, while the lower chambers contained RPMI-1640 with either PBS or PAUF. After 20.5 h, migrated cells were stained with Giemsa stain solution (Sigma-Aldrich, St. Louis, MO, USA) and counted. 

To evaluate the association between PAUF and TLR on monocyte chemotaxis, the cell suspension was treated with TLR2 inhibitor (TLR2-IN-C29, BioVision, Milpitas, CA, USA) and then loaded into the upper chambers. RPMI-1640 containing PBS or PAUF was loaded into the lower chambers. RPMI medium containing PAUF was added to the lower chambers, and Pam3Cys (Calbiochem, San Diego, CA, USA) and were used as a positive control. After 22.5 h, the cells that were moved were stained and captured using a microscope (Ni Eclipse, Nikon, Melville, NY, USA) with 100× magnification.

### 3.10. Cytokine Measurement

The cytokine production was quantified by the ELISA (enzyme-linked immunosorbent assay) method. Human TNF-α (#DY210-05, R&D systems, Minneapolis, MN, USA) and IL-10 (#ab46034, Abcam, Cambridge, UK) ELISA kits were used according to manufacturer’s instructions. The microplate reader (Versamax, Molecular Devices, San Jose, CA, USA) measured absorbance at 450 nm, and the cytokine concentration was determined using specific standard curves. 

### 3.11. Arginase Activity Assay

The Arginase Activity Assay Kit (#ab180877, Abcam, Cambridge, UK) was used to determine arginase activity in differentiated macrophages, following the manufacturer’s instructions. Briefly, cells were lysed in lysis buffer followed by the addition of arginase substrate mix and incubated at 37 °C for 20 min. After incubation, the reaction mixture was added and incubated at 37 °C for 30 min, and then optical density was measured by multimode plate reader, VICTOR Nivo 3F (PerkinElmer, Waltham, MA, USA) at 570 nm. Arginase activity was compared with an arginine standard curve. 

### 3.12. Reactive Oxygen Species (ROS) Measurement

Carboxy-H2DCFDA (#C400, Invitrogen, Waltham, MA, USA) was used to stain macrophages at a concentration of 10 μM for 1 h at 37 °C, 5% CO_2_. After incubation, cells were washed in PBS. Cells were suspended in phenol red-free RPMI 1640 (#11835030, Gibco, Waltham, Massachusetts, USA) at 37 °C for 15 min and then washed with cold PBS. The NovoCyte flow cytometer (Agilent Technologies, Santa Clara, CA, USA) analyzed fluorescence signals at an excitation/emission of 490/520 nm.

### 3.13. Immunofluorescence Analysis

Prior to monocyte seeding, cover slides were treated with a 0.1% gelatin coating (Sigma-Aldrich, St. Louis, MO, USA) in PBS. Adherent cells were incubated at 37 °C for 7 days, followed by washing with PBS and fixation in pre-cooled 4% paraformaldehyde at room temperature for 10 min. Next, the slides were rinsed two times with 10% FBS in PBS and then treated with 1% BSA in PBS for 1 h. Staining for CD86 and CD206 was performed using CD86-PE (#557344, BD Biosciences) or CD206-APC (#550889, BD Biosciences) for 1 h at 4 °C. After incubation, slides were washed twice with PBST (0.05% Tween20, Sigma-Aldrich). For visualization, slides were stained with DAPI (Sigma-Aldrich, St. Louis, MO, USA) at 37 °C for 30 min. The slides were prepared with antifade mounting medium (#H-1000, Vector Laboratories, Burlingame, CA, USA) and fluorescence images were captured using a Nikon fluorescence microscope (Ni Eclipse, Melville, NY, USA).

### 3.14. Western Blot Analysis

To measure the inducible nitric oxide synthase (iNOS) expression in the culture medium, differentiate macrophage culture medium was harvested. The culture medium was prepared in radioimmunoprecipitation assay (RIPA) buffer (50 mM Tris-Cl, 150 mM NaCl, 1% sodium deoxycholate (#D6750, Sigma-Aldrich, St. Louis, MO, USA), 5 mM EDTA (#E5134, Sigma-Aldrich, St. Louis, MO, USA), 30 mM Na_2_HPO_4_, 50 mM NaF, 1 mM Na_3_VO_4_). The BCA protein assay kit (Thermo Fisher Scientific, Waltham, MA, USA) was used to determine protein concentrations. The samples underwent SDS-PAGE and were then transferred to a nitrocellulose membrane (GE Healthcare). The membranes were probed with anti-human iNOS-HRP (#ab3523, 1:1000, Abcam, Cambridge, UK). The Azure C300 gel imaging system (Azure Biosystems, Dublin, CA, USA) detected the band’s intensity.

### 3.15. T Cell Proliferation Assay

T cells were stained with CFSE (5(6)-Carboxyfluorescein diacetate N-succinimidyl ester) at a concentration of 5 μM, following the manufacturer’s instructions from Invitrogen (#C34554, Waltham, MA, USA). Cells were washed three times before use. Next, 2 × 10^4^ or 4 × 10^4^ or 8 × 10^4^ CFSE-labeled T cells were cultured with 4 × 10^4^ differentiated macrophages in round-bottom 96-well tissue culture plates for 4 days. T cells were activated by combining them with anti-CD3/CD28 microbeads (Dynabeads™ Human T-Activator CD3/CD28 for T Cell Expansion and Activation, #11161D, Gibco, Waltham, Massachusetts, USA) in a 1:1 ratio in RPMI complete medium. T cells and macrophages were co-cultured in direct contact. After that, suspension cells were collected and stained for CD4-PerCP (#317432, Biolegend, San Diego, CA, USA) and CD8-FITC (#555366, BD Biosciences) and analyzed by a flow cytometer (Novocte, Agilent Technologies, Santa Clara, CA, USA). Before flow cytometry, dynabeads and bead-bound cells were removed using pipetting and magnets.

### 3.16. Phagocytosis Analysis

Phagocytosis assays were performed using a Phagocytosis Assay Kit (#ab234053, Abcam). The macrophages were stained with Green Zymosan for 2 h at 37 °C, 5% CO_2_. After incubation, cells were rinsed twice in phagocytosis assay buffer. To visualize phagocytosis, the samples were stained with DAPI for 30 min at 37 °C. Mounting medium was used to mount the slides, and fluorescence images were captured with a Nikon fluorescence microscope (Ni Eclipse, Melville, NY, USA). 

### 3.17. Flow Cytometry

Cell surface markers from the cell suspensions were stained with various combinations of fluorescent-labeled antibodies. Extracellular markers were suspended in flow cytometry staining buffer (5% FBS, 0.02% sodium azide in PBS) and stained with a fluorescent antibody. Following cell fixation and permeabilization, intracellular markers were labeled using a fluorescent antibody. Cells were fixed with 4% formaldehyde for 15 min at room temperature and permeabilized with 0.3% Triton X-100 for 15 min at room temperature. The NovoCyte flow cytometer was used to analyze the percentage of positive cells, which determined the marker expression. NovoExpress software version 1.5.0 (Agilent Technologies, Santa Clara, CA, USA) was used to analyze the data. 

### 3.18. Statistical Analysis

Experiments were repeated at least three times for statistical analysis. The assays were conducted on triplicate samples. Statistical differences were assessed using either a two-tailed paired Student’s *t*-test or a one-way analysis of variance (ANOVA) test for multiple comparisons. The data were expressed as mean  ±  standard deviation (SD). Statistically significant results were defined as *p* values < 0.05. GraphPad Prism version 8.0 (GraphPad Software, San Diego, CA, USA) was used to analyze the experimental data. 

## 4. Conclusions

In recent years, researchers have described several strategies for targeting macrophages in anti-cancer treatment [37], including (1) depleting macrophages systemically (e.g., using bisphosphonate), (2) inhibiting the recruitment or differentiation of macrophage precursors (e.g., using M-CSF/CSF-1 or CSF-1R inhibitors), and (3) reprogramming macrophages to exhibit an M1-like phenotype (e.g., using CD40 agonists). Since PAUF has functional similarity to M-CSF, anti-PAUF antibodies can play a role similar to that of CSF-1 inhibitors. The CSF-1 receptor is targeted by Cabiralizumab to inhibit the activation and survival of monocytes and macrophages [38]. The combination therapy of Cabiralizumab and nivolumab did not outperform nivolumab alone in treating advanced PC, as per phase 2 clinical trial results [38]. These results suggest that targeting multiple stromal cells in the TME with antibodies could be a more effective approach for cancer treatment compared to solely inhibiting immune cells. We intend to examine the effects of an anti-PAUF antibody on inhibiting TAM polarization and compare it to macrophage-targeted treatments.

## Figures and Tables

**Figure 1 ijms-25-11545-f001:**
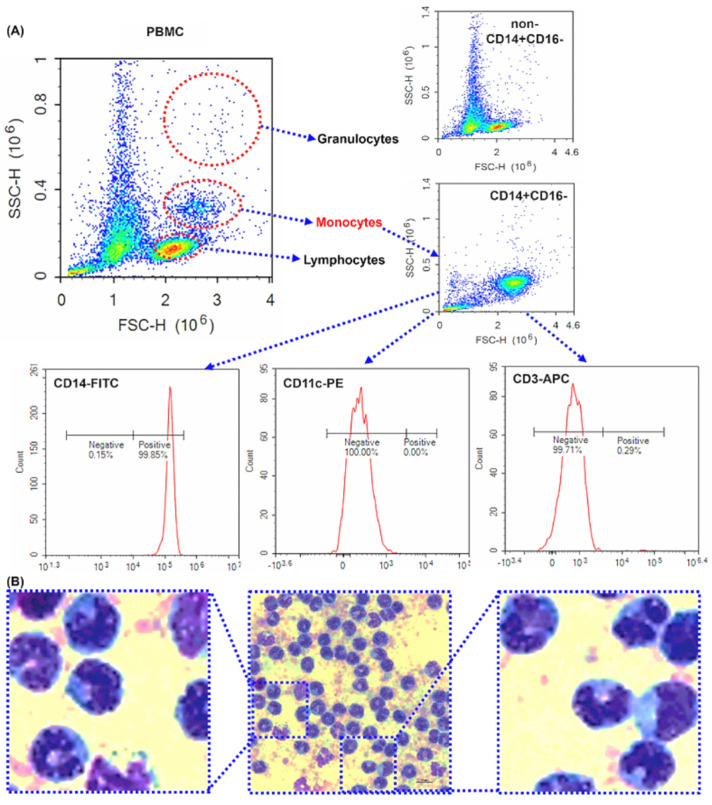
The purity of isolated human peripheral blood CD14^+^CD16^−^ monocytes—Purity investigation of magnetically isolated CD14^+^ monocytes by flow cytometry and microscope. Data were analyzed using NovoExpress software version 1.5.0. (**A**) The dot plots of forward scatter against side scatter. Representative dot plot of human PBMCs, CD14^−^, and CD14^+^ cells. CD14^+^ cells were surface labeled with CD14-FITC, CD11c-PE, CD3-APC antibodies. (**B**) Images representative of the subject were taken using a microscope. Monocytes were stained with Wright-Giemsa, revealing dark purple nuclei and a sky blue cytoplasm. Scale bar is 10 μm. 1000× magnification.

**Figure 2 ijms-25-11545-f002:**
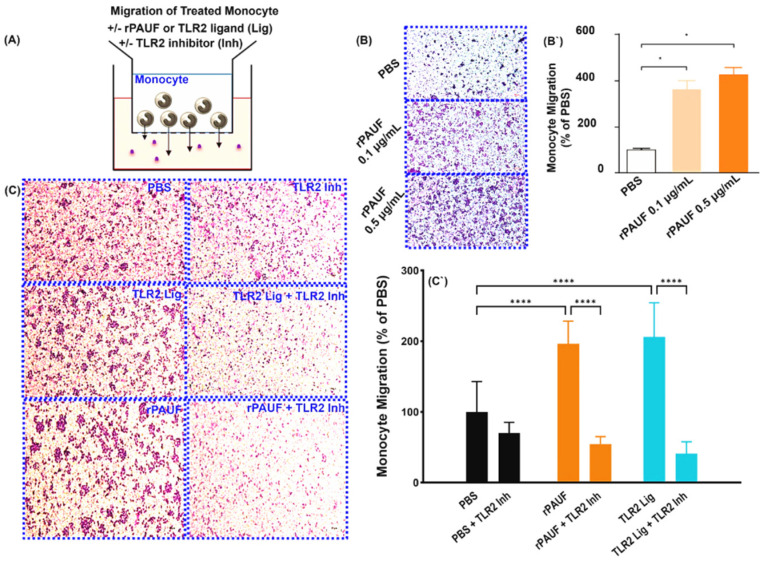
PAUF induces the chemotactic migration of monocytes—Chemotaxis of monocytes by transwell migration assay. (**A**) Schematic of the transwell chemotaxis model. Monocytes were seeded in the upper chamber at 3 × 10^5^ cells/200 μL. The chemoattractant PAUF (0.1 μg/mL or 0.5 μg/mL) was added to the lower chamber. PBS served as a control. (**B**) After 20.5 h of incubation, the migrating monocytes were stained with Giemsa stain, and the cells in the fields were counted. (**B`**) The bar chart displayed the percentage of migrated cells compared to the PBS group. (**C**) Human CD14^+^ monocytes were seeded in the upper chamber at 1.5 × 10^5^ cells/200 μL with or without TLR2 inhibitor (125 μM). The lower chamber was supplemented with the chemoattractant PAUF (0.5 μg/mL), Pam 3Cys (100 ng/mL). After 22.5 h of incubation, Giemsa stain was used to stain the migrated monocytes and the cells were counted in captured fields. Representative images of PBS and PAUF treated groups. All scale bars are 200 μm. 100× magnification. (**C`**) Bar chart showing the number of migrated cells, represented as a relative percentage to PBS group. The data were presented as the mean ± standard deviation from a minimum of three independent experiments. *, *p* < 0.05; ****, *p* < 0.0001.

**Figure 3 ijms-25-11545-f003:**
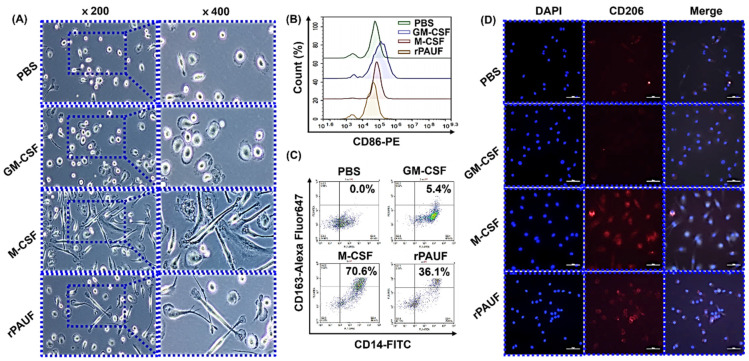
PAUF-MDMs exhibit morphological similarities to M2 macrophages—Healthy donor monocytes were isolated and differentiated into macrophages using GM-CSF (50 ng/mL), M-CSF (50 ng/mL), or PAUF (1 μg/mL) for 7 days in RPMI-1640 with 10% FBS. The resulting macrophages were categorized as M1 (GM-MDM), M2 (M-MDM), or PAUF-MDM. (**A**) Macrophage morphology was confirmed by a microscope. Scale bars are 50 μm. 200×, 400× magnification. MDMs were stained with (**B**) CD86-PE or (**C**) CD163-Alexa Fluor 647 antibody and analyzed using flow cytometry at day 7. (**D**) Fluorescence analysis of M2 marker CD206-APC (red) and nucleus (blue, DAPI) in differentiated macrophages. Scale bars are 50 μm. 400× magnification.

**Figure 4 ijms-25-11545-f004:**
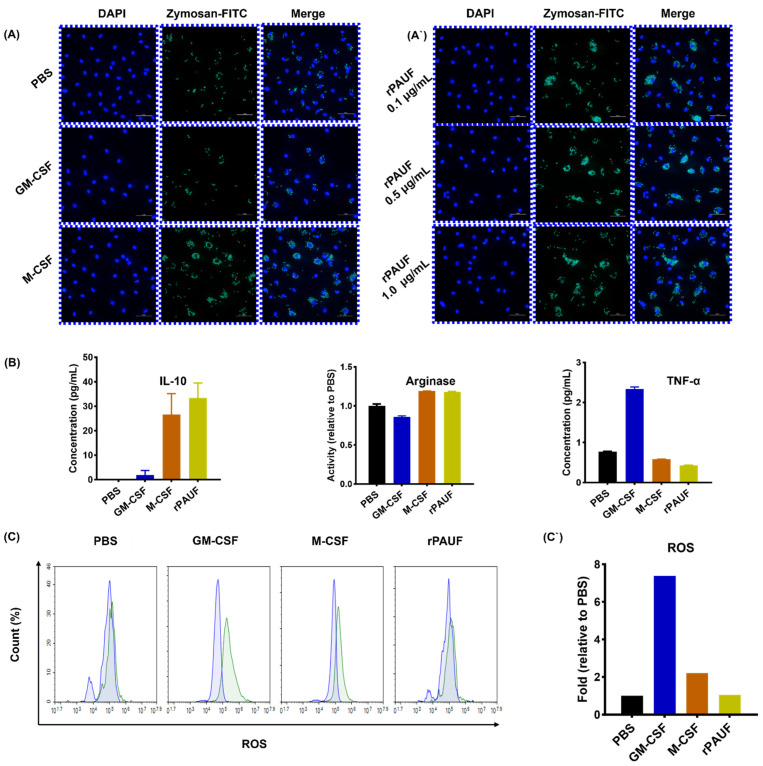
PAUF increases the phagocytosis ability of macrophages—(**A**) Fluorescence microscopy images of macrophages exposed to Zymosan for 2 h and stained with DAPI (nuclear stain). Cells were seeded on 0.1% gelatin-coated cover slides and differentiated for 7 days, followed by Zymosan reaction, and then fluorescence images were confirmed. Scale bars are 50 μm. 400× magnification. Blue, DAPI-stained nuclei; Green, fluorescent Zymosan. (**A`**) Images were taken in the presence of rPAUF (0.1 µg/mL, 0.5 µg/mL and 1 μg/mL). (**B**) Assessment of macrophage polarization was determined by comparison of the marker expression of M1 (TNF-α) and M2 (IL-10 and Arginase). On day 7, the levels of TNF-α and IL-10 were analyzed using ELISA in cell culture supernatants. Arginase activity was measured by a colorimetric assay. (**C**) The intracellular ROS production was analyzed by flow cytometry using Carboxy-H2DCFDA. The blue represents the cell-only group, while the green indicates ROS labeled with FITC. (**C`**) The bar chart represented a relative percentage of the PBS group. The data were presented as the mean ± SD from a minimum of three independent experiments.

**Figure 5 ijms-25-11545-f005:**
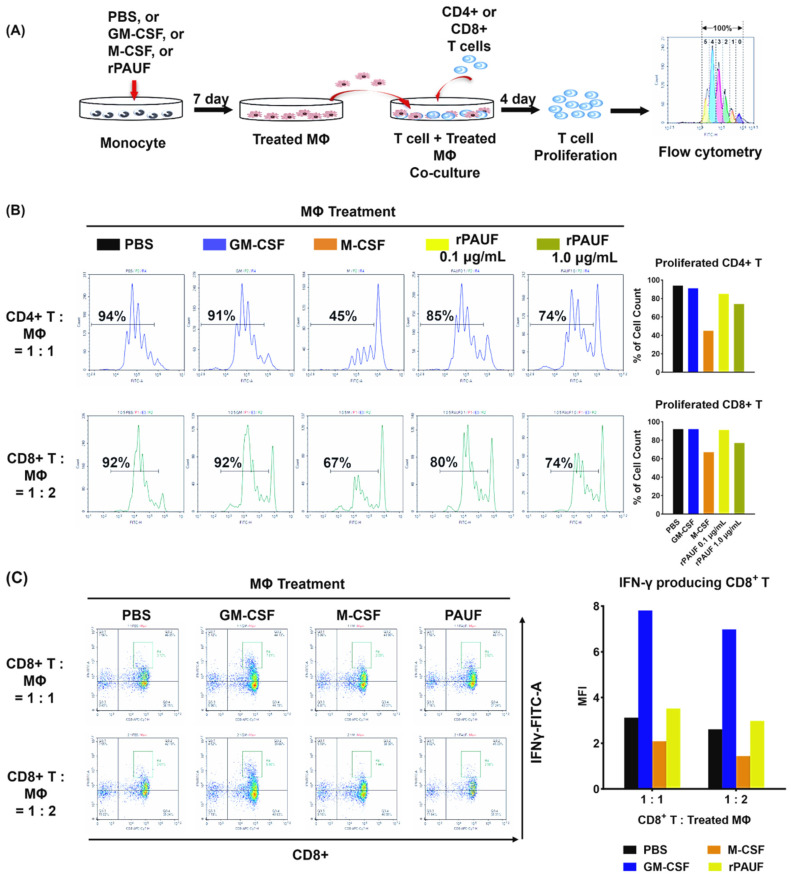
PAUF promotes the transformation of monocytes into M2 macrophages by inhibiting T cell proliferation. (**A**) Schematic description of the experiments. Monocytes differentiated into macrophages by incubating with PBS, GM-CSF, M-CSF, or PAUF for 7 days. To perform the suppression assay, isolated (**B**) CD4^+^ T cells or (**C**) CFSE-labeled CD8^+^ T cells were stimulated with anti-CD3/CD28 microbeads and co-cultured with macrophages in 1:1 or 1:2, or 1:0.5 ratio, respectively. After 4 days of co-culture, T cells were gathered and marked with anti-CD4-PerCP or CD8-PE/Cyanine7 to assess T cell growth utilizing flow cytometry. Data were analyzed using NovoExpress software version 1.5.0. The histograms represent the CFSE fluorescent peaks of proliferating T cell populations.

## Data Availability

The original contributions presented in the study are included in the article, further inquiries can be directed to the corresponding authors.

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
