# Peer review of "Pancreatic Adenocarcinoma Up-Regulated Factor (PAUF) Transforms Human Monocytes into Alternative M2 Macrophages with Immunosuppressive Action"

_ijms, 2024, doi:10.3390/ijms252111545_

Round 1
Reviewer 1 Report
Comments and Suggestions for Authors
Research article entitled Pancreatic Adenocarcinoma Up-regulated Factor (PAUF) trans- 2 forms Human Monocytes into Alternative M2 Macrophages 3 with Immunosuppressive Action was well received.
In this article, the authors have demonstrated the immunosuppressive role of PAUF on macrophage polarization and its subsequent effects on T-cell functions.
The article is well written. The study design looks appropriate. The results are clearly presented. However, here are some suggestions for the authors to consider.
Line 77-79…. Revise and rephrase. Not giving the intended meaning.
Please clarify if PAUF directly affected T cell functions or indirectly via modulating TAMs. If indirectly, please be more specific in the discussion part where the role of PAUF on T cell functions has been mentioned vaguely.
Author Response
Reviewer -1: Research article entitled Pancreatic Adenocarcinoma Up-regulated Factor (PAUF) trans- 2 forms Human Monocytes into Alternative M2 Macrophages 3 with Immunosuppressive Action was well received.
In this article, the authors have demonstrated the immunosuppressive role of PAUF on macrophage polarization and its subsequent effects on T-cell functions.
The article is well written. The study design looks appropriate. The results are clearly presented. However, here are some suggestions for the authors to consider.
Line 77-79…. Revise and rephrase. Not giving the intended meaning.
Answer: Thank you. We are grateful for your feedback. We have revised the manuscript based on your valuable comment. On page 2, you can find information about the revisions.
Before revision:
The relationship between PAUF and TAMs was explored using primary human monocytes. Healthy donor blood was used to obtain PBMCs, from which CD14-positive monocytes were isolated using magnetic beads. The purity of the isolated monocytes was analyzed by flow cytometric analysis (Figure 1A) and microscopic (Figure 1B) of the expression of monocyte, dendritic cell, and lymphocyte pan-surface markers, CD14, CD11c, and CD3, respectively. The purity of CD14-positive monocytes was >95%.
After revision:
Primary human monocytes were used to investigate the relationship between PAUF and TAMs. Magnetic beads were used to isolate CD14-positive monocytes from PBMCs obtained from healthy donor blood. The purity of the isolated monocytes was assessed through flow cytometric analysis (Figure 1A). Microscopic examination (Figure 1B) of CD14, CD11c, and CD3 pan-surface markers reveals expression of monocyte, dendritic cell, and lymphocyte. CD14-positive monocytes had a purity of more than 95%.
Please clarify if PAUF directly affected T cell functions or indirectly via modulating TAMs. If indirectly, please be more specific in the discussion part where the role of PAUF on T cell functions has been mentioned vaguely.
Answer: On page 9, you can find information about the revisions.
After revision:
The study suggests that PAUF-induced M2-like macrophages can maintain antitumor properties, challenging the traditional polarization concept in tumor microenvironments. The impact of PAUF on TAMs was shown in their ability to suppress T cell functions, including CD4+ T cell proliferation and CD8+ T cell activity. The results indicate that PAUF indirectly influences T cells by modulating their functions through TAM-mediated mechanisms in the TME.
Reviewer 2 Report
Comments and Suggestions for Authors
It is demonstrated that Pancreatic adenocarcinoma upregulated factor (PAUF) functions as a promoter of cancer progression by regulating the recruitment and differentiation of macrophages within the tumor microenvironment (TME), ultimately causing immunosuppression.
In the introduction, you wrote: TAMs promote the proliferation of T-helper 2 (Th2) and not Th1 cells by producing pro-inflammatory cytokines and activate regulatory T cells to induce immune tolerance. TAMs promote angiogenesis by producing anti-inflammatory cytokines and support the invasive and metastatic abilities of tumor cells.
Please clearly describe what you want to express.
Figure 5: Please indicate the number of experiments and, if possible, the SD. How did you determine that the shape and position of the gate were correct?
Author Response
Reviewer-2: It is demonstrated that Pancreatic adenocarcinoma upregulated factor (PAUF) functions as a promoter of cancer progression by regulating the recruitment and differentiation of macrophages within the tumor microenvironment (TME), ultimately causing immunosuppression.
In the introduction, you wrote: TAMs promote the proliferation of T-helper 2 (Th2) and not Th1 cells by producing pro-inflammatory cytokines and activate regulatory T cells to induce immune tolerance. TAMs promote angiogenesis by producing anti-inflammatory cytokines and support the invasive and metastatic abilities of tumor cells.
Please clearly describe what you want to express.
Answer: Thank you. We are grateful for your feedback. We have revised the manuscript based on your valuable comment. On page 2, you can find information about the revisions.
After revision:
The presence of TAMs in tumors leads to tumor progression via the secretion of growth factors, cytokines, and other signaling molecules. They stimulate the growth of T-helper 2 (Th2) cells, trigger Tregs to promote immune tolerance, and hinder Th1-mediated immune responses. The production of anti-inflammatory cytokines by TAMs promotes angiogenesis and enhances the invasive and metastatic capabilities of tumor cells. TAMs have been studied as possible targets for therapy to inhibit tumor growth and spread due to their significant role in the tumor microenvironment.
Figure 5: Please indicate the number of experiments and, if possible, the SD. How did you determine that the shape and position of the gate were correct?
Answer: Figure 5 displays the results of FACS analysis, with three samples combined for each condition. To ensure accurate analysis, we combined cells from three separate experiments due to the limited number of cells in each sample. The figure does not display the individual experimental variation or standard deviation for these specific experiments. The same analysis was performed using the GATE method stored in the device.